# A Histology-Guided Approach to the Management of Patients with Lupus Nephritis: Are We There Yet?

**DOI:** 10.3390/biomedicines10061409

**Published:** 2022-06-15

**Authors:** Bogdan Obrișcă, Alexandra Vornicu, Alexandru Procop, Vlad Herlea, George Terinte-Balcan, Mihaela Gherghiceanu, Gener Ismail

**Affiliations:** 1Department of Nephrology, Fundeni Clinical Institute, 022328 Bucharest, Romania; bogdan.obrisca@drd.umfcd.ro (B.O.); vornicu.alexandra@yahoo.com (A.V.); 2Department of Nephrology, “Carol Davila” University of Medicine and Pharmacy, 020021 Bucharest, Romania; 3Department of Pathology, Fundeni Clinical Institute, 022328 Bucharest, Romania; procop_alex@yahoo.com (A.P.); herlea2002@yahoo.com (V.H.); 4Ultrastructural Pathology, “Victor Babes” National Institute of Pathology, 050097 Bucharest, Romania; george.terinte@gmail.com (G.T.-B.); mgherghiceanu@yahoo.com (M.G.)

**Keywords:** lupus nephritis, kidney biopsy, repeat biopsy, protocol biopsy, biomarkers, histology

## Abstract

Renal involvement is a frequent complication of systemic lupus erythematosus (SLE). It occurs in up to two-thirds of patients, often early during the disease course, and is the most important predictor of the morbidity and mortality of SLE patients. Despite tremendous improvements in the approach of the lupus nephritis (LN) therapy, including the recent approval of two new disease-modifying therapies, up to 50% of patients do not obtain a renal response and up to 25% will eventually progress to end-stage renal disease (ESRD) within 10 years of diagnosis. Given the lack of correlation between clinical features and histological lesions, there is an increasing need for a histology-guided approach to the management of patients with LN. Apart from the initial diagnosis of type and severity of renal injury in SLE, the concept of a repeat kidney biopsy (either in a for-cause or a per-protocol scenario) has begun to gain increasing popularity in the nephrology community. Herein, we will provide a comprehensive overview of the most important areas of utility of the kidney biopsy in patients with LN.

## 1. Introduction

Systemic lupus erythematosus (SLE) is the prototype of autoimmune disorders caused by a loss of tolerance to endogenous nuclear antigens triggering an aberrant autoimmune response that determines a broad spectrum of organ manifestations with a wide range of severity [1,2]. Renal involvement occurs in up to two-thirds of patients with SLE, often early during the disease course, and is the most important contributor to the morbidity and mortality of these patients as a consequence of disease activity, target organ damage, and adverse events associated with aggressive immunosuppressive (IS) therapy [1,3].

Recent years have brought ground-breaking results to the LN treatment landscape that have led to the FDA-approval of two new diseases—modifying therapies (belimumab and voclosporin) as add-on agents to the current standard of care (mycophenolate mofetil or cyclophosphamide-based regimens) [4,5]. Despite this important progress, the complete renal response rates with these newer treatment regimens do not exceed 50% after two years of therapy, leaving an unacceptable high number of patients for whom there is still need for further improvement [6]. In addition, these therapies may pose a high-cost burden despite their potential clinical superiority, which may augment the uncertainty regarding their optimal clinical use and cost-effectiveness [7].

In line with the overall low-response rates, up to 30% of patients with LN will eventually progress to ESRD [8]. In a meta-analysis of 187 articles that reported the renal outcome on 18.309 patients, the 10-year and 15-year ESRD risk showed a stepper decline in the 1970s and 1980s, plateaued in mid 1990s, and then showed a notable increase in the late 2000s, especially in those with class IV LN [9]. The reason for this renal outcome evolution of patients with LN may be due to the recent shifts in treatment away from high-dose cyclophosphamide and high-dose steroid regimens, which may actually offer less control over disease activity in LN [9].

In fact, one possible limitation of the current LN trials is that they fail to identify the highest risk patients with the most severe underlying histology that may actually benefit from different treatment regimens. It is now increasingly acknowledged that clinical and histologic features of LN can be discordant [10]. The past clinical trials that evaluated the high-dose cyclophosphamide NIH (National Institutes of Health) regimens had adequate and detailed reporting of underlying histologic activity and chronicity, which allowed to identify the patients most likely to respond to IS therapy [11,12]. By contrast, the recent trials focused on continuous refinement of the clinical response criteria such as proteinuria threshold, the magnitude of eGFR (estimated glomerular filtration rate) decline from baseline, and the activity of urinary sediment or steroid doses at different time points [10]. Despite the fact that this approach may work in the clinical trial setting, the different definitions employed make the direct comparison of results between trials difficult.

Accordingly, the one-size-fits-all approach may not be applicable in a clinical setting when an individualized patient management is needed. As such, the kidney biopsy remains an invaluable tool to guide the management of patients with LN, but there is an unmet need to better refine the indications for the kidney biopsy in patients with SLE, to define the value of initial histology in the management of such patients and to identify the clinical scenarios in which a repeat biopsy is needed.

Herein, we will provide a comprehensive overview of the most important areas of utility of the kidney biopsy in patients with LN.

## 2. Renal Involvement in SLE

During the past decades, attention has focused on patients with LN in terms of a better understanding of the pathogenesis, the characterization of the long-term outcomes, and the refinement of the IS regimens. Nonetheless, we need to acknowledge that virtually all renal compartments can be affected in SLE, with different patterns of injury frequently overlapping, complicating the assessment of prognosis and obligating to an individualization of the patient’s management (Table 1).

### 2.1. Glomerular Lesions in SLE

To date, there are no clinical features, laboratory tests, or imaging modalities capable of predicting the histology of the kidney in LN [10]. Apart from the classical LN, several other patterns of glomerular injury have been described in patients with SLE with distinct pathogenic mechanisms.

Crescents are commonly encountered in biopsy specimens of patients with LN, while the incidence, pathogenic mechanism, and clinical outcome of true crescentic necrotizing glomerulonephritis (GN) is less well described [15]. In a study examining the histologic characteristics of 105 patients with LN, crescents were encountered in 64% of kidney biopsies, while 12% of patients had pathological features of crescentic GN with more than 50% of the examined glomeruli with crescents [17]. In our experience, crescents were encountered in 37% of biopsy samples, and the mean percentage of glomeruli showing extracapillary hypercellularity was 10.7 ± 23% [8]. In a study of 152 patients with class IV-G LN, 21.7% (n = 33) had true crescentic GN. All patients had acute kidney injury at diagnosis, higher activity and chronicity scores on pathological examination, and poorer long-term renal outcomes, with fewer patients achieving a complete remission and more progressing to ESRD [18]. Despite the fact that most cases of patients with crescentic GN are classified as a diffuse proliferative LN (IV-G), implying that the mechanism of capillary wall necrosis and crescent formation is a consequence of the intensity and severity of immune complex deposition and of complement activation, forms of crescentic GN that are reminiscent of ANCA-associated vasculitis can be encountered. In a study comparing 254 LN biopsies, of whom 32 were ANCA-positive and 222 LN were ANCA-negative, it was shown that ANCA-positivity was associated with a distinct clinical and pathological phenotype of LN [16]. As such, these patients more frequently had glomerular necrosis and segmental lesions (were classified more frequently as class IV-S), had higher serum creatinine at biopsy, and had higher immunological activity (higher anti-dsDNA ab titers and lower serum C4 concentrations) [16]. Similarly, other studies have identified lower levels of immunoglobulins and complement on immunofluorescence in crescentic LN compared to non-crescentic LN, further supporting a common pathogenic mechanism with ANCA-associated vasculitis [18].

Renal manifestations of SLE can be a consequence of injury to every cell type, including podocytes. Most frequently, podocyte injury occurs subsequent to immune complex deposition in glomerular capillary walls. Nonetheless, there is accumulating evidence of podocyte injury in SLE that morphologically resembles minimal-change disease (MCD) or focal and segmental glomerulosclerosis (FSGS) with diffuse foot process effacement (≥70%), with mesangial dense deposits and the absence of any subepithelial or subendothelial dense deposits [14]. These observations led to the introduction of the term “lupus podocytopathy” to describe this distinct clinical entity. A study examined the clinical and morphological features of 50 patients with lupus podocytopathy (1.33% of 3750 biopsies from patients with SLE) and showed that all patients presented with nephrotic syndrome [14]. In addition, 34% had acute-kidney injury at diagnosis and frequently presented with moderate-severe acute tubulo-interstitial injury. Moreover, the majority (94%) achieved a response (complete or partial remission) after a median 4–8 weeks of IS therapy. This is in contrast with the median time to remission observed in LN (7–9 months, up to 18 months) and further supports the notion that lupus podocytopathy is a distinct clinical entity [19].

In addition to lupus podocytopathy, a newly recognized pathological entity called podocyte infolding glomerulopathy is consistently reported in association with SLE. It is characterized by the appearance of numerous microspheres or microtubular structures in the glomerular basement membrane (GBM) believed to be invaginations of podocytes occurring as a result of intra-GBM complement activation [20].

### 2.2. Tubulointerstitial Lesions in SLE

Despite the fact that the current ISN/RPS (International Society of Nephrology/Renal Pathology Society) classification focuses primarily on glomerular pathology, it is recommended that both tubulointerstitial and vascular lesions should be described in the pathology report [21]. Moreover, there are kidney biopsies showing a discordance between the severity of tubulointerstitial or vascular lesions and that of glomerular pathology, thus suggesting a different pathogenic mechanism [15]. In a study of 313 patients with LN, the scores of interstitial inflammatory cell infiltration, tubular atrophy, and interstitial fibrosis were most severe in class IV, moderate in class III, and mild in classes II and V [22]. Among these, a subgroup of 15 patients with no or mild glomerular lesions had severe tubulo-interstitial lesions and distinct clinical features, with more patients presenting with anemia and fewer with hematuria compared to the other subgroups [22]. Additionally, tubulointerstitial lesions were found to be independent predictors of renal outcome after multivariate adjustment (interstitial inflammation, HR 1.84 (95% CI, 1.22–2.77); tubular atrophy, HR 2.35 (95% CI, 1.01–2.93); and interstitial fibrosis, HR 1.95 (95% CI, 1.23–2.23)) [22]. Similarly, we have identified tubulitis as being independently associated with a 13.1-fold higher risk of a worse outcome (HR, 13.1; 95% CI, 1.3–131) [8].

Apart from these tubulointerstitial lesions, the presence of interstitial B-cells organized into tertiary lymphoid structures are being increasingly recognized as a driver of local autoimmunity and inflammation [1]. Moreover, the presence of these structures correlates with the severity of both glomerular and tubulointerstitial lesions. As such, the efficacy of various IS agents on improving the clinical outcome in LN might be related to their capacity to deplete interstitial B-cells [1,23,24,25,26].

### 2.3. Vascular Lesions in SLE

Vascular lesions are frequently encountered in patients with SLE and, despite the fact that they have received little attention over the past decades, their presence might be associated with renal outcome, patient survival, and vascular events [15]. In a retrospective analysis of 429 kidney biopsies from patients with SLE, 44% had arterial sclerosis, 1.4% had non-inflammatory necrotizing vasculopathy, 5.4% had thrombotic microangiopathy (TMA), and 2.6% had true renal vasculitis [27]. Those with TMA or true renal vasculitis had worse renal function at baseline and higher blood pressure. Another study identified that the presence of renal vascular lesions was associated with arterial vascular events (including myocardial infarction, angina, cerebrovascular accident, and transient ischemic attacks) [13]. In another study of 197 patients with LN, 25.4% had co-existing renal TMA [28]. These patients had more severe clinicopathological features compared to patients without coexistent TMA, with higher rates of oliguria, more advanced kidney injury, and more extensive fibrocellular/fibrous crescents and tubular atrophy. Moreover, their prognosis was worse, with lower rates of clinical remission and higher rates of treatment failure and death [28].

## 3. The Role of Initial Kidney Biopsy in the Management of Patients with SLE and Renal Involvement—Beyond the ISN/RPN Lupus Nephritis Classification

### 3.1. When to Perform a Kidney Biopsy in Patients with SLE? Is the Current Biopsy Threshold Adequate?

Kidney biopsy remains the gold-standard in assessing the renal involvement in SLE, evaluating the severity of underlying lesions, and, thus, guiding the management of such patients. Despite the fact that all the major clinical trials in LN used clinical endpoints (based on proteinuria, urinary sediment, and/or eGFR decline) to assess the treatment efficacy, there is in fact a weak correlation between clinical features and histological findings in patients with LN [10]. There is an increasing need to incorporate adequate histological data into clinical trials, both in terms of inclusion criteria and the assessment of treatment response.

The available clinical guidelines have discordant indications on when to perform a kidney biopsy in patients with SLE. The 2012 American College of Rheumatology Guideline for LN indicates that to perform a kidney biopsy in patients with a decline of renal function, a proteinuria over 1 g/day or over 0.5 g/day with additional abnormalities of urinary sediment [29]. Similarly, the joint report of EULAR/ERA-EDTA (European League Against Rheumatism/European Renal Association-European Dialysis and Transplant Association) recommendations for the management of LN and the recently updated KDIGO 2021 Guideline (Kidney Disease: Improving Global Outcomes) for the management of glomerular disease retain the proteinuria threshold of 500 mg/day (with or without abnormal urinary sediment) and the decline in renal function as indications for kidney biopsy in patients with SLE [30,31].

Nonetheless, there is substantial evidence from observational studies that patients without clinical signs of renal involvement have substantial histologic activity (Table 2, Figure 1). In a study of 195 patients with SLE, 86 patients had no clinical signs of renal involvement and available renal pathology [32]. A proliferative LN (class III or IV) was found in 15% of these patients. Recently, De Rosa et al. compared 46 patients with SLE and a proteinuria less than 0.5 g/day to 176 patients with a proteinuria level over 0.5 g/day [33]. Despite the fact that the frequency of proliferative classes of LN and the activity/chronicity index were significantly higher among those with proteinuria over 0.5 g/day, the presence of low-level proteinuria did not exclude the possibility of a severe underlying histology [33]. Among those with low-level proteinuria, a proliferative class of LN was encountered in approximately 85% of patients, while the median activity index was 6 (and up to 14). This shows the importance of keeping a low-level threshold for kidney biopsy indication in order to adequately identify all patients that might benefit from IS therapy.

### 3.2. Interpretation of the Initial Biopsy Information beyond the Lupus Nephritis Classifications

Apart from initial diagnosis of renal involvement in SLE, the histologic evaluation of kidney biopsies remains essential to further guide the therapeutic approach. Since the initial publication of the World Health Organization (WHO) classification of LN in 1974, over the past decades there have been several revisions of the original form, including a transition to the ISN/RPS LN classification in 2003 (Table 3).

Despite the fact that several studies have shown that these classifications are able to predict the long-term outcome of LN, there are still a number of criticisms on its ability to guide treatment and forecast LN prognosis [21,38,39,40,41]. Given the histologic heterogeneity of renal involvement in SLE and the wrong assumption that the prognosis of patients within the same class of LN is the same irrespective of the histological lesions encountered, it became obvious that the concept of “one-size-fits-all” is not suitable for LN. Accordingly, continuous efforts need to be undertaken to better refine the histological approach to this disorder [8]. First of all, the initial transition from the WHO 1982 to the ISN/RPS 2003 LN classification led to the introduction of the segmental and global subdivisions of the class IV LN [37]. Despite the observations that there are clinical and pathological differences between class IV-S and IV-G, the former behaving as a pauci-immune GN and the latter as an immune complex GN, and a tendency towards a worse outcome for patients with class IV-S, a meta-analysis has concluded that the renal outcomes of these two subdivisions are similar [42,43,44,45]. This led to the elimination of segmental and global subdivisions of class IV in the 2018 revision of the ISN/RPS LN classification [21]. Nonetheless, the segmental-global debate is not entirely without any prognostic implications, and, in fact, it is possible that the transition from the WHO 1982 to the ISN/RPS 2003 classification may have led to an “artificial” attenuation of the prognosis of the patients with the most severe underlying histology (Table 3). This was shown in a study by Schwartz et al., in which 39 biopsies with WHO class IV and 44 with WHO III ≥ 50% LN were reclassified using the ISN/RPS classification [46]. Patients with severe segmental GN (WHO III ≥ 50%) have segmental lesions in over 50% of the non-sclerotic glomeruli and a worse outcome than patients with diffuse global GN (WHO class IV), possibly behaving like a pauci-immune vasculitis [40]. When reclassified by the ISN/RPS classification, of the 44 patients with WHO III ≥ 50%, 22 patients had class IV-S LN and 22 patients transferred to class IV-G LN (and further named class IV-Q). The class IV-Q patients had a worse rate of survival at 10 years than patients with class IV-S and those with “true” class IV-G (WHO class IV) [46]. In fact, the reclassification according to the ISN/RPS criteria in only two subdivisions of class IV led to a dramatic attenuation of the prognosis of those patients with the actual worse survival. These observations further highlight the limitations of the ISN/RPS classification. Thus, there is a need to better refine the histological assessment of patients with LN and to rely on an objective and quantitative assessment of histologic lesions rather than on an arbitrary approach, in a manner similar to the Banff Classification of renal allograft pathology [47].

In an attempt to overcome the limitations of the ISN/RPS classification, Rijnink et al. analyzed the prognostic significance for renal outcome in LN of 50 histological variables outside the framework of the ISN/RPS classification [17]. ESRD was predicted by both histological variables (fibrinoid necrosis (HR, 1.08 per %; 95% CI, 1.02–1.13), fibrous crescents (HR, 1.09 per %; 95% CI, 1.02–1.17), and interstitial fibrosis/tubular atrophy in over 25% of the cortical area (HR, 3.89; 95% CI, 1.25 to 12.14)) and clinical variables (eGFR at baseline (HR, 0.98 per ml/min per 1.73 m^2^; 95% CI, 0.97–1.00), and non-white race (HR, 7.16; 95% CI, 2.34–21.91)) [17]. In an attempt to validate this approach, we identified that, in addition to eGFR (HR, 0.91 per ml/min per 1.73 m^2^, 95% CI, 0.85–0.91) and 24-h proteinuria (HR, 2.04 per g/day; 95% CI, 1.19–3.5), the percentage of glomeruli with crescents (HR, 1.06 per %; 95% CI, 1.003–1.13), the presence of adhesions (HR, 9.2; 95% CI, 1.38–61.2), and the presence of tubulitis (HR, 13.1; 95% CI, 1.3–131) are independent predictors of outcome in patients with LN [8].

Thus, the 2018 revision of the ISN/RPS classification proposed that the designation of activity/chronicity by the terms A, A/C, and C is too broad and should be replaced by a semiquantitative assessment of histological lesions in the form of a modified NIH activity and chronicity index [21,48]. Nonetheless, despite an overall improvement to the previous approach, the NIH indices should be further validated in independent cohorts and should be implemented in the design of clinical trials [49]. In addition, as these modified scores do not incorporate all the histologic lesions encountered in patients with SLE, the biopsy report should at least describe the presence of additional lesions as these might be related to outcome and might potentially be a treatment modifier [21].

## 4. Role of Repeat Kidney Biopsy

The role of repeat kidney biopsy has been increasingly recognized over the past decades as an invaluable tool to assess the histological stratification and adjust immunosuppressive therapy in patients with LN [2]. Nonetheless, despite the fact that several scenarios in relation to the moment of the repeat biopsy have been described, there is little consensus on the optimal approach [50]. The first scenario is for-cause repeat kidney biopsy performed in the following settings: persistent proteinuria (to differentiate persistent histologic activity from scarring), the progressive increase of serum creatinine or treatment unresponsiveness, and flare rebiopsy [50]. The second scenario is per-protocol repeat kidney biopsy, post-induction therapy (to assess treatment response), and during maintenance therapy (to assess the optimal moment for IS withdrawal) [50].

### 4.1. Post-Induction Therapy Repeat Kidney Biopsy

Several studies over the past decades have evaluated the utility of per-protocol repeat biopsy post-induction therapy (Table 4). Malvar et al. evaluated 69 consecutive patients that underwent a repeat biopsy after a 6-month period of induction therapy (consisting of high-dose corticosteroids and either monthly intravenous cyclophosphamide or mycophenolate mofetil) [51]. Overall, there was a significant improvement of the activity index (AI) between the two biopsies, but among those with a complete renal response only 50% had an AI of 3 or less, while 29% had an AI of 5 or higher. In addition, among patients that achieved a complete histologic remission (AI of 0), 62% showed a residual proteinuria over 500 mg/day. These results highlight the discrepancy between the clinical and histological remission in LN and, in the face of divergent clinical trials end-points, the incorporation of histological end-points might increase the strength of RCTs (randomized clinical trials) and better delineate which patients would benefit the most from different treatment regimens.

A collaborative effort from the Lupus Nephritis Trials Network designed a prospective, randomized, multinational study with the aim to investigate whether a per-protocol repeat kidney biopsy after 12 months from initiation of IS therapy in incident cases of active LN results in treatment changes and, subsequently, in improved long-term outcomes (ReBioLup study, “Per-protocol repeat kidney biopsy in incident cases of lupus nephritis”, ClinicalTrials.gov Identifier: NCT04449991) [2]. Similarly, we are currently undertaking a randomized, clinical trial with the aim to compare the efficacy of two IS regimens (EUROLUPUS vs. RITUXILUP regimen [52,53]) with complete histological remission (AI of 0) defined as the primary end-point (GLUREDLUP study, “Minimizing Glucocorticoid Administration in Patients With Proliferative Lupus Nephritis”, ClinicalTrials.gov Identifier: NCT05207358).

### 4.2. During Maintenance Therapy Repeat Kidney Biopsy

A per-protocol repeat kidney biopsy during maintenance therapy is another strategy to evaluate the optimal moment for IS therapy withdrawal (Table 5).

De Rosa et al. evaluated, by a repeat kidney biopsy, 36 patients that stopped the IS therapy after being in complete clinical remission for at least 12 months and having received at least 36 months of IS [74]. About a third of patients (n = 11) had a flare during follow-up, of whom the majority (n = 10) had residual histologic activity. All patients with an AI of 2 or higher at the second biopsy flared, while the presence of endocapillary hypercellularity was among the strongest predictors of a future LN flare [74]. In another study, 75 patients were prospectively managed by repeated kidney biopsies during maintenance therapy [78]. These patients received at least 42 months of immunosuppression (with least 12 months of complete clinical remission) and had the maintenance therapy withdrawn only if the repeat biopsy showed a complete histologic remission (AI of 0), while the IS therapy was continued if the AI was 1 or higher [78]. This approach to the maintenance therapy management was effective, with only seven patients flaring during a follow-up period of 50 months (flare rate of 1.5/year), much less than the rate previously reported [78]. In addition, at the third kidney biopsy, only 29% of patients with an AI of 0 had a complete negative immunofluorescence examination, further stressing the incomplete clearance of immune complexes from the glomeruli that pose a risk on future LN relapses [78]. As such, there is need to better incorporate the histologic information gained from the kidney biopsy in the management of patients with SLE and renal involvement (Figure 2).

## 5. Potential Biomarkers Reflecting the Activity of Lupus Nephritis

Given that the kidney biopsy remains an invasive procedure with potential serious complications, there is a need to identify and validate potential biomarkers reflecting the disease activity in LN [10,81]. Several biomarkers have been evaluated over the past decades (reviewed in [10,82]), but none has gained acceptance to penetrate into clinical practice (Table 6).

Complement activation plays a major role in the pathogenesis of LN and is the mediator of renal injury subsequent to immune complex deposition [83]. The intensity of membrane attack complex (C5b-9) deposition in the tubular basement membrane correlated with the severity of tubulointerstitial damage, while the intensity of C5b-9 capillary wall deposition was associated with non-response to IS therapy [84]. Nonetheless, despite the fact that the evaluation of complement activation might be valuable in assessing the underlying activity of LN, the quantification of C5b-9 deposition still needs a kidney biopsy.

One of the most promising non-invasive tools to assess disease activity in LN is the measurement of urinary soluble CD163 (usCD163) [86]. CD163 is a transmembrane protein functioning as a scavenger receptor for the hemoglobin-haptoglobin complexes and is expressed mainly on activated M2c macrophages [10]. The soluble CD163 is presumed to be shed in the urine by intrarenal M2c macrophages and has been evaluated as a biomarker reflecting histologic activity in ANCA-associated vasculitis, IgA nephropathy, and LN [86,87,88,89]. In LN, the level of usCD163 correlated with histologic class and the histologic activity index, whereas a repetitive assessment showed that the level increased with up to 6 months before a LN flare, decreased in responders, and remained elevated in non-responders [86]. Despite the fact that it requires validation in independent cohorts and, ideally, in a clinical trial setting, the possibility of repetitive and quantitative assessments of usCD163 make it a potential dynamic indicator of underlying disease activity and tissue injury in LN.

In summary, kidney biopsy remains an invaluable tool to guide the management of patients with LN. Given the unsatisfactory renal response rates with current immunosuppressive regimens in LN, there is a need to better stratify the risk of progression and incorporate tissue-based information into meaningful clinical trials endpoints.

## Figures and Tables

**Figure 1 biomedicines-10-01409-f001:**
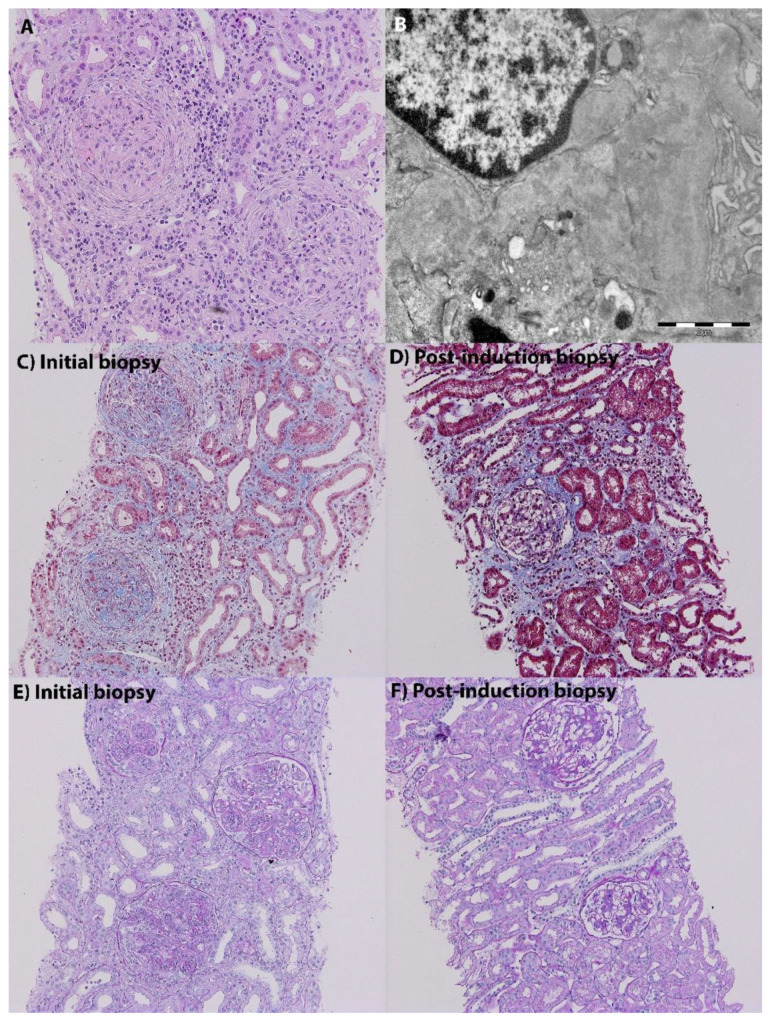
Patient 1 (**A**,**B**). Patient with SLE with normal renal function, 24-h proteinuria of 0.2 g/day, and minimal microscopic hematuria. Kidney biopsy shows severe, proliferative LN with crescent formation, and an activity index of 14 ((**A**), haematoxylin eosin staining, magnification 20×), with extensive mesangial and subendothelial immune complex deposition ((**B**), electron microscopy, magnification 11,000×). Patient 2, post-induction repeat biopsy (EUROLUPUS regimen). Initial biopsy, (**C**) (Masson staining, magnification 20×) and (**E**) (PAS staining, magnification 20×), show severe, proliferative LN with an activity index of 16 at baseline. Repeat biopsy, (**D**) (Masson staining, magnification 20×) and (**F**) (PAS staining, magnification 20×), show a significant histologic improvement with a decrease of the activity index to 2. (Images from the Renal Biopsy Registry of Fundeni Clinical Institute, Bucharest, Romania).

**Figure 2 biomedicines-10-01409-f002:**
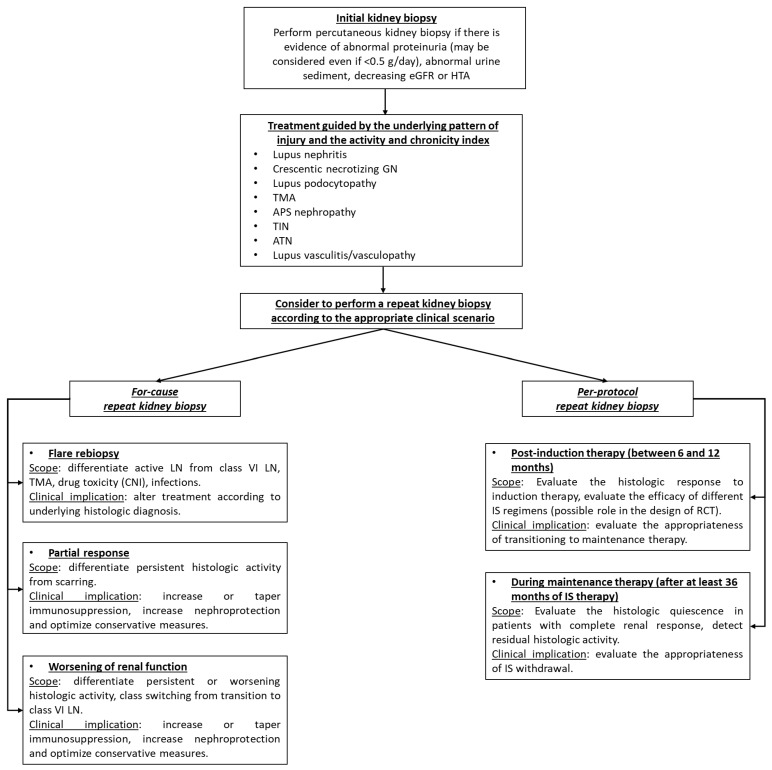
The integration of percutaneous kidney biopsy in the management of patients with SLE and renal involvement (adapted after [2,10,30,31,50,79,80]) (abbreviations: GN, glomerulonephritis; TMA, thrombotic microangiopathy; TIN, tubulo-interstitial nephritis; ATN, acute tubular necrosis; APS, anti-phospholipid syndrome; LN, lupus nephritis; CNI, calcineurin inhibitor; IS, immunosuppression; and RCT, randomized clinical trial).

**Table 1 biomedicines-10-01409-t001:** The types of renal injury in SLE (adapted after [13,14,15,16]).

Renal Compartment Involved	Clinical Context and Histological Aspect
(I) Glomerular compartment	
Lupus nephritis	Mesangial and subendothelial immune complex deposits (±subepithelial) associated with mesangial, endocapillary, and extracapillary hypercellularity. In the setting of pure membranous LN, there are mainly subepithelial immune complex deposits that may be accompanied by mesangial deposits and hypercellularity.
Crescentic necrotizing GN	Can occur in the setting of proliferative lupus nephritis. There are situations when there is a discrepancy between the magnitude of immune complex deposition (resembling a “pauciimune” appearance) and the severity of necrotizing lesions (frequently associated with ANCA positivity and possibly sharing some common pathogenic mechanism).
Lupus podocytopathy	A glomerular pattern of injury that is similar to MCD/FSGS with extensive foot process effacement (>70%). Can associate mesangial immune deposits and hypercellularity but NO subendothelial or subepithelial immune complex deposition.
Thrombotic microangiopathy	Associated with anti-phospholipid syndrome, TTP/HUS, or malignant HTA, or with an overlap with systemic sclerosis. Generalized endothelial dysfunction (endotheliosis), thrombi formation in small vessels (glomeruli and/or arterioles), the widening of subendothelial space, and mesangiolysis.
**(II) Tubulo-interstitial compartment**	
Tubulo-interstitial nephritis	Usually, tubulo-interstitial injury correlates with glomerular involvement, but, in rare cases, an isolated tubulo-interstitial nephritis can be encountered.
Tubulitis	Lymphocyte infiltration between the tubular basement membrane and the basolateral membrane of tubular epithelial cells; granular IgG immune complex deposition at this site.
Proximal tubular epithelial cells vacuolization	Intracytoplasmic vacuolization of tubular epithelial cells usually associated with massive proteinuria.
Acute tubular necrosis	Associated with massive proteinuria and/or red blood cell casts.
Tubular atrophy and interstitial fibrosis	Chronic, irreversible lesions as a consequence of active glomerular, tubulo-interstitial, or vascular lesions.
**(III) Vascular compartment**	
Lupus vasculopathy	Necrotizing changes in the vessel wall associated with abundant immune deposits causing luminal narrowing or occlusion. There is often positivity in immunofluorescence for fibrin, immunoglobulin, and complement with the absence of inflammatory cells.
Thrombotic microangiopathy	Generalized endothelial dysfunction (endotheliosis), thrombi formation in small vessels (glomeruli and/or arterioles), the widening of subendothelial space, and mesangiolysis. Histologically, it is identical to TTP/HUS lesions.
True renal vasculitis	The involvement of the small- and medium-sized arteries; there is a prominent inflammatory cell infiltrate with mural inflammation and fibrinoid necrosis resembling microscopic polyangiitis.
Uncomplicated vascular immune deposits—UVIDs	Lesions with vascular immune deposits that, when visualized by light microscopy, reveal that, despite the vessels appearing normal, immune complex deposits are present in the walls of arterioles and to a lesser extent, in the veins. No thrombosis or inflammatory infiltrate is present, and immunofluorescence is positive for immunoglobulins and complement.
Arteriosclerosis (AS)	The thickening of the medial layer of the interstitial arteries and/or arteriolar hyalinosis.

Abbreviations: UVIDs, uncomplicated vascular immune deposits; AS, arteriosclerosis; GN, glomerulonephritis; MCD, minimal change-disease; FSGS, focal and segmental glomerulosclerosis; TTP, thrombotic thrombocytopenic purpura; HUS, hemolytic uremic syndrome; ANCA, antineutrophil cytoplasmic antibodies; LN, lupus nephritis; and HTA, arterial hypertension.

**Table 2 biomedicines-10-01409-t002:** Studies evaluating the role of kidney biopsy in patients with low-level proteinuria.

Study	Number of pts.	Creatinine at Biopsy	Proteinuria at Biopsy	Hematuria at Biopsy	Class of LN	Mean AI and CI
**Mavragani (2015)** [34]	297	Creatinine >1.2 mg/dL -Cls. II: 23.3% -Cls. III/IV: 33.3% -Cls. V: 11.1%	***<0.25 mg/day*** -Cls. II: 15.9% -Cls. III/IV: 7.3% -Cls. V: 3.3% ***0.25–0.50 mg/day*** -Cls. II: 22.7% -Cls. III/IV: 16.8% -Cls. V: 15%	-Cls. II: 45.4% -Cls. III/IV: 75.1% -Cls V: 38.9%	Cls. II: 47 pt. Cls. III/IV: 188 pt. Cls: V: 62 pt.	NR
**Wakasugi (2012)** [32]	86 (no clinical signs)	0.6 mg/dL (0.3–1.0)	0 (0–350) mg/day	No pt. with active urinary sediment	Cls. I: 25 pt. Cls. II: 28 pt. Cls: III ± V: 8 pt. Cls: IV ± V: 5 pt. Cls: V: 9 pt.	NR
**Zabaleta-Lanz (2006)** [35]	30 (silent LN)	CrCl: 96.08 ± 17.78 mL/min	140 ± 80.7 mg/day	Normal urinary sediment	Cls. I: 2 pt. Cls. II: 19 pt. Cls. III: 6 pt. Cls. IV: 1 pt. Cls.: V: 2 pt.	AI: 2.9 ± 1.2 CI: 1.9 ± 1
**Chedid (2020)** [36]	87	*-Isolated low-level proteinuria* (52 pts.): 0.7 ± 0.2 mg/dL *-Low-level proteinuria with AKI ± µ hem*. (35 pts.) 1.5 ± 1.1 mg/dL	*-Isolated low-level proteinuria:* 0.6 ± 0.2 g/day *-Low-level proteinuria with AKI ± µ hem.* (35 pts.) 0.5 ± 0.2 g/day	*-Low-level proteinuria with AKI ± µ hem.* 51% of pts. with µ hem.	*-Isolated low-level proteinuria:* Cls. I/II—23% Cls. III/IV/V—53% *-Low-level proteinuria with AKI ± µ hem.* Cls. I/II—20% Cls. III/IV/V—62%	*-Isolated low-level* *proteinuria:* AI: 4.5 ± 2.1 CI: 2.7 ± 2.5 *-Low-level proteinuria with AKI ± µ hem.* AI: 5.5 ± 2.4 CI: 2.1 ± 2.4
**De Rosa (2020)** [33]	46	0.7 mg/dL (0.4–1.3)	Proteinuria: <0.5 g/day Median: 0.23 g/day (0–0.42)	All had glomerular hematuria	Cls. II: 10.9% Cls. III: 30.4% Cls. IV: 45.7% Cls. V: 4.3% Cls. III–IV + V: 8.7% Cls. VI: 0%	AI: 6 (0–14) CI: 2 (0–4)

Abbreviations: LN, lupus nephritis; AI, activity index; CI, chronicity index; Cls, class; NR, not reported; AKI, acute kidney injury; CrCl, creatinine clearence; and µ. hem., microscopic hematuria.

**Table 3 biomedicines-10-01409-t003:** The evolution of LN classifications (adapted after [21,37,38]).

	WHO 1974	WHO 1982	ISN/RPS 2003	ISN/RPS 2018
**Class I**	**Normal glomeruli**	**Normal glomeruli** a. Nil (by LM/IF/EM) b. Normal by LM, but deposits by IF/EM	**Minimal mesangial LN** Normal by LM, mesangial deposits by IF/EM	**Minimal mesangial LN** Normal by LM, mesangial deposits by IF/EM
**Class II**	**Purely mesangial disease** a. Normocellular mesangium by LM but mesangial deposits by IF/EM b. Mesangial hypercellularity with mesangial deposits	**Pure mesangial alterations** a. Mild hypercellularity b. Moderate hypercellularity	**Mesangial proliferative LN** Mesangial hypercellularity with mesangial deposits by IF/EM	**Mesangial proliferative LN** Mesangial hypercellularity with mesangial deposits by IF/EM
**Class III**	**Focal proliferative GN (<50%)**	**Focal segmental GN** a. With “active” necrotizing lesion b. With “active” and sclerosing lesions c. With sclerosing lesions	**Focal LN (<50%)** Class III (A) Class III (A/C) Class III (C)	**Focal LN (<50%)** Modified NIH lupus nephritis activity and chronicity scoring system to be used instead of the A, C, and A/C parameters
**Class IV**	**Diffuse proliferative GN (≥50%)**	**Diffuse GN** a. Without segmental lesions b. With “active” necrotizing lesion c. With “active” and sclerosing lesions d. With sclerosing lesions	**Diffuse LN (≥50%)** Class IV-S (A) Class IV-G (A) Class IV-S (A/C) Class IV-G (A/C) Class IV-S (C) ClassIV-G (C)	**Diffuse LN (≥50%)** Elimination of S and G subdivisions Modified NIH lupus nephritis activity and chronicity scoring system to be used instead of the A, C, and A/C parameters
**Class V**	**Membranous GN**	**Diffuse membranous GN** a. Pure membranous GN b. Associated with lesions of class II c. Associated with lesions of class III d. Associated with lesions of class IV	**Membranous LN**	**Membranous LN**
**Class VI**	**Not defined**	**Advanced sclerosing GN**	**Advanced sclerosing LN**	**Advanced sclerosing LN**

Abbreviations: LN, lupus nephritis; LM, light microscopy; IF, immunofluorescence; EM, electron microscopy; GN, glomerulonephritis; NIH, National Institutes of Health; A, active; C, chronic; A/C, active/chronic; WHO, World Health Organization; and ISN/RPS, International Society of Nephrology/Renal Pathology Society.

**Table 4 biomedicines-10-01409-t004:** Studies evaluating the role of repeat kidney biopsy in LN post-induction therapy.

Author (Year)	Nr. of pts.	Interval from 1st to 2nd Biopsy (mo)	Indications to Repeat Biopsy	Proteinuria at 1st and 2nd Biopsy (g/24 h)	Class of LN at 1st Biopsy	AI at 1st and 2nd Biopsy (Mean)	CI at 1st and 2nd Biopsy (Mean)
**Gunnarsson (2002)** [54]	18	6	Protocol	1st: 1.6 (0–19.8) 2nd: 0.5 (0–3.1)	III-7 pts. IV-11 pts.	1st: 8 (4–13) 2nd: 4 (0–13)	1st: 1 (0–4) 2nd: 2 (0–4)
**Hill (2001)** [55]	71	6	Protocol	NR	III-9 pts. IV-55 pts. III + V-7 pts.	1st: AI ≤ 10–16 pts. and >10–29 pt. 2nd: AI ≤ 4–29 pts. and >4 22 pts., ≤1–12 pts. and >6–15 pts.	1st: CI ≤ 2–28 pts. and >2–17 pts. 2nd: CI ≤ 2.5–24 pts., >2.5–27 pts.
**Askenazi (2007)** [56]	25 (ped. pop.)	9	Protocol	1st: 3.2 ± 2.6 2nd: 0.6 ± 0.8 (*p* < 0.002)	IV ± V-25 pts.	1st: A-68% and A/C-32% 2nd: A-53% and A/C-29%	1st: C-0% 2nd: C-18%
**Grootscholten (2007)** [57]	39	24	Protocol	1st: 3.6 (2.6–7.1) 2nd: 0.2 (0.1–2.2)	III-2 pts. IV-34 pts.	1st: 8.0 (6.0–12.0) 2nd: 2.7 (2–3.3)	1st: 2.7 (2.0–3.3) 2nd: 3.3 (2.7–4.7)
**Gunnarsson (2007)** [58]	7	3–12	Protocol	1st: 2.7 (0.2–5.9) 2nd: 0.8 (0.1–1.8)	III-1 pts. IV-6 pts.	1st: mean 6.42 2nd: mean 2.57	1st: mean 4 2nd: mean 4.14
**Wang (2008)** [59]	13	6	Protocol	NR for the pts. with repeat biopsies	IV-10/13 III-3/13	1st: 8.9 2nd: 2.2	1st: 0.8 2nd: 2.8
**Zickert (2014)** [60]	67	8 (5–15)	Protocol	1st: 1.4 (0–8.4) 2nd: 0.5 (0–3.6)	III-21 pts. IV-27 pts. III-IV/V-9 pts. V-10 pts.	1st: 5 (0–13) 2nd: 2 (0–12)	1st: 1 (0–6) 2nd: 1.5 (0–8)
**Singh (2014)** [61]	40	6	Protocol	1st: 2.5 ± 1.8 2nd: 0.9 ± 1.1	IV-70%	1st: 6.05 ± 2.9 2nd: 2.5 ± 2.5	1st: 0.68 ± 1.23 2nd: 2.52 ± 2.9
**Malvar (2017)** [51]	69	6.6 ± 0.7	Protocol	1st: 2.9 ± 2.1 2nd: 1.1 ± 1.3	III-20 pts. IV-49 pts.	1st: 8.5 ± 3.1 2nd: 3.5 ± 2.4	1st: 2.6 ± 1.7 2nd: 4 ± 1.5
**Tannor (2018)** [62]	31	6.4 (6.0–7.9)	Protocol	NR	24/31 pts.—prolif. class	1st: 7 (4–9) 2nd: 2 (1–4)	1st: 2.7 ± 1.7 2nd: 3.7 ± 1.6

Abbreviations: LN, lupus nephritis; AI, activity index; CI, chronicity index; Cls, class; NR, not reported; pts, patients; mo, months; ped. pop., pediatric population; A, active; C, chronic; A/C, active/chronic; pts., patients; and prolif., proliferative.

**Table 5 biomedicines-10-01409-t005:** Studies evaluating the role of repeat kidney biopsy in LN during maintenance therapy.

Author (Year)	Nr. of pts.	Interval from 1st to 2nd Biopsy (mo)	Indications to Repeat Biopsy	Proteinuria at 1st and 2nd Biopsy (g/24 h)	Class of LN at 1st Biopsy	AI at 1st and 2nd Biopsy (Mean)	CI at 1st and 2nd Biopsy (Mean)
**Esdaile (1993)** [63]	42	25	Protocol	1st: 0.99 2nd: 0.5	II-2 pts. III-4 pts. IV-31 pts. V-5 pts.	1st: 7 2nd: 2	1st: 2 2nd: 2
**Yoo (2000)** [64]	21	43 ± 31	Clinical/ Protocol	*Pts. with clinical progression*: 1st: 2.9 ± 1.2 2nd: 2.1 ± 1.2 *Pts. without clinical progression*: 1st: 1.3 ± 0.8 2nd: 2.5 ± 3.4	IV-21 pts.	*Pts. with clinical progression*: 1st: 2.9 ± 1.2 2nd: 2.1 ± 1.1 *Pts. without clinical* *progression*: 1st: 1.3 ± 0.8 2nd: 1.5 ± 0.8	GS(%) *Pts. with clinical progression*: 1st: 5.1 ± 7.1 2nd: 49 ± 29.4 *Pts. without clinical progression*: 1st: 1.7 ± 3 2nd: 8.9 ± 10.4
**Huraib (2000)** [65]	21	24	Protocol	1st: 2.81 ± 2.4 2nd: 1.39 ± 1.5	IV-17 pts. V-4 pts.	1st: 10.7 ± 3.6 2nd: 7.8 ± 3.3	1st: 3.2 ± 1.9 2nd: 6.3 ± 3.5
**Zhang (2009)** [66]	31	12	Protocol	1st: 4.8 ± 2.7 2nd: 1.8 ± 1.2	II-1 pt. III-11 pts. IV-10 pts. V-9 pts.	1st: 12.6 ± 5.8 2nd: 4.8 ± 2.1	1st: 2.4 ± 1.5 2nd: 2.6 ± 1.8
**Stoenoiu (2011)** [67]	30	24 ± 6	Protocol	*AZA group*: 1st: 3.3 ± 2.8 2nd: 0.5 ± 1.1 *MMF group*: 1st: 3.5 ± 3.0 2nd: 0.6 ± 1.1	*AZA group* IV ± V-10 pts. *MMF group* IV ± V-11 pts.	*AZA group* 1st: 10 (3–14) 2nd: 2 (0–14) *MMF group* 1st: 8.5 (5–16) 2nd: 3.5 (0–9)	*AZA group* 1st: 1 (0–3) 2nd: 2.5 (0–5) *MMF group* 1st: 1 (0–3) 2nd: 2.5 (1–7)
**Wang (2012)** [68]	44	NR	Clinical/ Protocol	1st: 3.0 ± 1.8 2nd: 2.8 ± 2.1	II-5 pts. III-4 pts. IV-22 pts. V-3 pts. III/IV + V-16 pts.	1st: 5.8 ± 3.0 2nd: 4.7 ± 2.6	1st: 1.8 ± 1.2 2nd: 3.4 ± 2.0
**Alsuwaida (2012)** [69]	77	12–18	Protocol	1st: 1.3 (0.53–3.8) 2nd: N/A	II-8 pts. III-27 pts. IV 28 pts. V-7 pts. III/IV + V-6 pts. VI-1 pts.	*Entire cohort* 1st: 3 (1–9) *Pts. with CR* 1st: 2 (1–9) 2nd: 1 (0–2) *Pts. with PR* 1st: 3 (1–9) 2nd: 2 (0–3) *Pts. with NR* 1st: 4 (0–8) 2nd: 3 (1–9)	*Entire cohort* 1st: 3 (2–5) *Pts. with CR* 1st: 2.5 (2–4.5) 2nd: 4 (2–7) *Pts. with PR* 1st: 4 (2–6) 2nd: 5 (2–6) *Pts. with NR* 1st: 3 (2–5) 2nd: 6 (5–7)
**Alsuwaida (2013)** [70]	11 pts. with 3 serial biopsies each	1st–2nd: 24 mo 2nd–3rd: 42 mo	Clinical	1st: 1.1 ± 0.8 2nd: 1.6 ± 1.4 3rd: 2.6 ± 1.9	II-3 pts. III-1 pt. IV-6 pts. V-1 pt.	1st: 3.1 ± 4.2 2nd: 5 ± 4.3 3rd: 4.9 ± 4.9	1st: 2.5 ± 2.5 2nd: 5.8 ± 2.3 3rd: 5.3 ± 2.9
**Pagni (2013)** [71]	142	4.9 years (±4.9)	Clinical/ Protocol	1st: 3.5 ± 3.9 2nd: 3.1 ± 3.1	II-18 pts. III-15 pts. IV-72 pts. V-24 pts. Mixed-13 pts.	1st: 4.5 ± 3.8 2nd: 3.3 ± 3.3	1st: 1.5 ± 1.8 2nd: 3.6 ± 2.7
**Alvarado (2014)** [72]	25	2nd: 6 3rd: at least 42 mo	Protocol	1st: 3.3 ± 2.09 2nd: 1.1 ± 0.7 3rd: 0.3 ± 0.2	N/A	1st: 8.9 ± 4.1 2nd: 4.3 ± 2.7 3rd: 0.96 ± 1.2	1st: 2.8 ± 1.4 2nd: 4.2 ± 1.8 3rd: 4.3 ± 1.6
**Pineiro (2016)** [73]	35	30 ± 9	Clinical	1st: 4.1 ± 2.8 2nd: 0.6 ± 1.1	III and IV-33 pts. IV + V-2 pts.	1st: 9.9 ± 3.4 2nd: 1.3 ± 1.9	1st: 1.5 ± 1.6 2nd: 2.4 ± 1.7
**De Rosa (2018)** [74]	36	min. 36 mo. of IS	Protocol	1st: 2.1 (0.2–20) 2nd: 0.11 (0.03–0.48)	III-13 pts. (+V-4/13) IV-23 pts.	1st: 8 (3–16) 2nd: 0 (0–5)	1st: 3 (0–6) 2nd: 3 (0–5)
**Parodis (2020)** [75]	42	24.3	Protocol	1st: 2.0 (1.0–3.5) 2nd: 0.2 (0.1–0.7)	III ± V-12 pts. IV ± V 30 pts.	1st: 8.5 (6.0–10.3) 2nd: 3.0 (1.0–4.3)	1st: 1.0 (0.0–3.0) 2nd: 2.0 (2.0–4.0)
**Morales (2021)** [76]	26	71 ± 10	Clinical	1st: 2.8 (1.1–4.31) 2nd: 2.83 (1.79–4.88)	II-8 pts. III-2 pts. IV-10 pts. V-3 pts. III/IV + V-3 pts.	1st: 2 (0–8.2) 2nd: 1 (0–4.5)	1st: 1 (0–2) 2nd:3 (2–4.2)
**Das (2021)** [77]	29	61 ± 18	Protocol	1st: 3.9 ± 2.1 2nd: 0.24 ± 0.1	III-3 pts. IV-25 pts. IV + V-1 pt.	1st: 8 (3–20) 2nd: 93.1% with AI of 0	1st: 1 (0–3) 2nd: 2 (0–3)

Abbreviations: LN, lupus nephritis; AI, activity index; CI, chronicity index; Cls, class; N/A, not available; pts, patients; mo, months; CR, complete response; PR, partial response; NR, no response; AZA, azathioprine; MMF, mycophenolate mofetil; and GS, glomerulosclerosis.

**Table 6 biomedicines-10-01409-t006:** Traditional and potential biomarkers reflecting LN activity (adapted after [10,82,85].

Serum Biomarkers	Urine Biomarkers
Serum creatinine Anti-dsDNA ab Anti-C1q ab Anti-nucleosome ab Serum C3 and C4 Interferon signature Blood neutrophil signature	Proteinuria Hematuria NGAL KIM-1 MCP-1 TWEAK VCAM-1 Osteoprotegerin IL-6/IL-8/IL-17 Transferrin Ceruloplasmin usCD163

Abbreviations: ab, antibodies; dsDNA, double-stranded DNA, IL, interleukin; NGAL, neutrophil gelatinase-associated lipocalin; KIM-1, kidney injury molecule 1; MCP-1, monocyte. chemoattractant protein 1; TWEAK, tumor necrosis factor-like inducer of apoptosis; and VCAM-1, vascular-cell adhesion molecule 1.

## Data Availability

Not applicable.

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
