# Peer review of "A Histology-Guided Approach to the Management of Patients with Lupus Nephritis: Are We There Yet?"

_biomedicines, 2022, doi:10.3390/biomedicines10061409_

Round 1

Reviewer 1 Report

The Authors did a terrific job assembling papers and trials about lupus nephritis.I do agree in the need to incorporate kidney biopsy and repeated kidney biopsy in patients affected by SLE not only to better define therapy management but also to drive further clinical trials.

In order to increase the already high quality of the paper, I would make a few suggestions:

1. Introduction.

The first five lines of the introduction are the same reported in the abstract: I would ask to change those a little bit.

2. Page 2, line 73

The start is not clear; please rearrange the sentence.

3. Page 8, Fig.1

Pictures should have a scale bar or the magnification reported in the description.

4. Page 10, lines 277-278

Did the Author mean WHO III <= 50% instead to >= 50%?

5. Page 15, Table V

Typing error, “su” instead to “us”.

Author Response

Dear Biomedicines Editorial team,

On behalf of the co-authors I want to thank you for the opportunity of incorporating editorial and reviewer comments made in relation to our manuscript entitled “A histology-guided approach to the management of patients with lupus nephritis: are we there yet?”. We hope to have addressed all the comments and suggestions and believe that it has made our manuscript more clear and meaningful for publication.

Sincerely,

Gener Ismail

Reviewer 1

The Authors did a terrific job assembling papers and trials about lupus nephritis. I do agree in the need to incorporate kidney biopsy and repeated kidney biopsy in patients affected by SLE not only to better define therapy management but also to drive further clinical trials.

Thank you for the appreciation and suggestions. We incorporated your suggestions and hope made this manuscript better.

In order to increase the already high quality of the paper, I would make a few suggestions:

  1. Introduction.

The first five lines of the introduction are the same reported in the abstract: I would ask to change those a little bit.

We have changed the first part of abstract.

  1. Page 2, line 73

The start is not clear; please rearrange the sentence.

We have adjusted the sentence according to your suggestions.

  1. Page 8, Fig.1

Pictures should have a scale bar or the magnification reported in the description.

We have added the magnifications of each figure. We have replaced the figure B with a figure adequately measured that has also a scale bar.

  1. Page 10, lines 277-278

Did the Author mean WHO III <= 50% instead to >= 50%?

The correct form is WHO >= 50%. This is a sub-entity defined in the WHO classification in patients with segmental lesions found in over 50% of the glomeruli.

  1. Page 15, Table V

Typing error, “su” instead to “us”.

Thank you for the observation. We have corrected the typing error.

Reviewer 2 Report

The authors of the manuscript “A histology-guided approach to the management of patients with lupus nephritis: are we there yet?” aimed to highlight the relevance of histological assessment in lupus nephritis (LN). The subject is interesting and relevant to clinicians, pathologists, and researchers.

The authors cite older relevant research and recent work on the subject, including articles published in 2021 and 2022, and the manuscript provides insight on the value of histological assessment in LN (classification, pre and post-induction results, kidney biopsy during maintenance therapy).

Whereas the manuscript is well documented, in its current form, some readers may find it hard to follow. In order to prevent this from happening, the general ideas presented in the article could be reorganized in a more systematic way (adding subchapters and sub-subchapters could be helpful) - so that the readers can find the information they need more easily.

Indeed, readers (including clinicians) who are interested in the subject would probably seek answers to the following questions:

·      -   What are the general characteristics of renal involvement in SLE? – clinical, immunological, biochemical findings (including biomarkers of activity presently included in the 5th section of the manuscript), and histopathologic findings (frequency, description of the lesions + a short description of the pathomechanism, technique, diagnostic value, classification, …). The information presented in Table 1 could be turned into sub-subchapters.

·     -    What is the value of the initial biopsy and when should it be performed? – recommendations, the value of performing a biopsy in patients with/without (or minimal) clinical manifestations (subchapters could be useful), and predictive value.

·     -    What is the value of the repeat biopsy and when should it be performed? – general indications, the correlation between the findings - and the response to different treatment regimens (induction/maintenance – these could be subchapters). Table 4 could be split into 2 shorter tables (and would be placed in the 2 subchapters).

·     -    How is the kidney biopsy integrated into the management of LN?

The abovementioned questions are answered (almost in full) by the contents of the article, yet specific relevant information is harder to find immediately in the current form. Please reorganize the information.

Moreover, most paragraphs are rather long. Please divide the information into smaller paragraphs (separate specific ideas/statements).

The manuscript’s introduction should mainly emphasize the importance of performing a kidney biopsy in LN (ex: value for diagnosis, follow-up, …) and highlight the currently unmet needs. Treatment options could be discussed elsewhere (ex: repeat biopsy findings – findings according to different treatment regimens).

L41: “After decades of negative trials in lupus nephritis, recent years brought ground-breaking results to the LN treatment landscape that led to the FDA approval of two new disease-modifying therapies (belimumab and voclosporin)” – this statement suggests that there are no other currently available treatment options for LN. Please rephrase.

Figure 1 is more suited for the section regarding repeat biopsies in LN.

L73: “Despite that, …” – ambiguous, please rephrase.

L174: “There is an increasingly need to incorporate…” - increasingly

Finally, please remember to define any acronyms before using them in the text. Additionally, please define the acronyms used in the tables (below the tables).

Overall, I believe that the manuscript should be reorganized in order to improve clarity.

Author Response

Dear Biomedicines Editorial team,

On behalf of the co-authors I want to thank you for the opportunity of incorporating editorial and reviewer comments made in relation to our manuscript entitled “A histology-guided approach to the management of patients with lupus nephritis: are we there yet?”. We hope to have addressed all the comments and suggestions and believe that it has made our manuscript more clear and meaningful for publication.

Sincerely,

Gener Ismail

Reviewer 2

The authors of the manuscript “A histology-guided approach to the management of patients with lupus nephritis: are we there yet?” aimed to highlight the relevance of histological assessment in lupus nephritis (LN). The subject is interesting and relevant to clinicians, pathologists, and researchers.

The authors cite older relevant research and recent work on the subject, including articles published in 2021 and 2022, and the manuscript provides insight on the value of histological assessment in LN (classification, pre and post-induction results, kidney biopsy during maintenance therapy).

Whereas the manuscript is well documented, in its current form, some readers may find it hard to follow. In order to prevent this from happening, the general ideas presented in the article could be reorganized in a more systematic way (adding subchapters and sub-subchapters could be helpful) - so that the readers can find the information they need more easily.

Indeed, readers (including clinicians) who are interested in the subject would probably seek answers to the following questions:

  • - What are the general characteristics of renal involvement in SLE? – clinical, immunological, biochemical findings (including biomarkers of activity presently included in the 5th section of the manuscript), and histopathologic findings (frequency, description of the lesions + a short description of the pathomechanism, technique, diagnostic value, classification, …). The information presented in Table 1 could be turned into sub-subchapters.

  • - What is the value of the initial biopsy and when should it be performed? – recommendations, the value of performing a biopsy in patients with/without (or minimal) clinical manifestations (subchapters could be useful), and predictive value.

  • - What is the value of the repeat biopsy and when should it be performed? – general indications, the correlation between the findings - and the response to different treatment regimens (induction/maintenance – these could be subchapters). Table 4 could be split into 2 shorter tables (and would be placed in the 2 subchapters).

  • - How is the kidney biopsy integrated into the management of LN?

The abovementioned questions are answered (almost in full) by the contents of the article, yet specific relevant information is harder to find immediately in the current form. Please reorganize the information.

Moreover, most paragraphs are rather long. Please divide the information into smaller paragraphs (separate specific ideas/statements).

The manuscript’s introduction should mainly emphasize the importance of performing a kidney biopsy in LN (ex: value for diagnosis, follow-up, …) and highlight the currently unmet needs. Treatment options could be discussed elsewhere (ex: repeat biopsy findings – findings according to different treatment regimens).

Thank you for the appreciation and constructive remarks. We have reorganized the manuscript in distinct subchapters in each specific are discussed (we have divided the chapter describing types of renal involvement in SLE, the chapters approaching the initial kidney biopsy interpretation and the one approaching the repeat kidney biopsy). We have also divided the table IV in two distinct tables so the reader follows more easily the data regarding the post-induction repeat biopsy studies and the data regarding the maintenance-repeat biopsy studies. In addition, we have readjusted some paragraphs to make them more easy to follow. Thank you for those inputs as we believe that improved the quality of the manuscript. We have also adjusted the introduction section.

L41: “After decades of negative trials in lupus nephritis, recent years brought ground-breaking results to the LN treatment landscape that led to the FDA approval of two new disease-modifying therapies (belimumab and voclosporin)” – this statement suggests that there are no other currently available treatment options for LN. Please rephrase.

Thank you for the suggestion. We have modified the phrase and hope its reflects better the current treatment regimens in LN.

Figure 1 is more suited for the section regarding repeat biopsies in LN.

Figure 1 contains both a patient that was biopsy with minimal urinary findings and another patient with a repeated biopsy post-induction therapy. As the journal guidelines request that we put the figure at the first reporting throughout the manuscript we chose this place.

L73: “Despite that, …” – ambiguous, please rephrase.

Thank you for the suggestions. We have rephrased this paragraph.

L174: “There is an increasingly need to incorporate…” - increasingly

We have corrected this sentence

Finally, please remember to define any acronyms before using them in the text. Additionally, please define the acronyms used in the tables (below the tables).

Thank you for the observation. Indeed, it was a missed aspect. We have added adequate abbreviations list below each table.

Overall, I believe that the manuscript should be reorganized in order to improve clarity.

Reviewer 3 Report

The paper is a review regarding the role of kidney biopsy in patients with lupus nephritis. The paper is interesting and sound. I have the following comments for the authors:

- Given that even in the first parts of section 2 the authors discussed the role of LN class, I suggest the authors to consider the possibility to modify the organization of the paper in order to firstly report the LN classification. 

- Please review the paper in order to define the term the first time they appear in the text (eg: page 1, line 42 - LN; page 2, line 49 - ESRD; page 4, line 90 - GN; page 5, line 149 - IS; page 5, line 172 - eGFR; page 5, line 180 EULAR/ERA-EDTA; page 5, line 182 - KDIGO). 

- Please report the legend for table I, II, III, IV and V. 

Author Response

Dear Biomedicines Editorial team,

On behalf of the co-authors I want to thank you for the opportunity of incorporating editorial and reviewer comments made in relation to our manuscript entitled “A histology-guided approach to the management of patients with lupus nephritis: are we there yet?”. We hope to have addressed all the comments and suggestions and believe that it has made our manuscript more clear and meaningful for publication.

Sincerely,

Gener Ismail

Reviewer 3

The paper is a review regarding the role of kidney biopsy in patients with lupus nephritis. The paper is interesting and sound. I have the following comments for the authors:

Thank you for the appreciation and suggestions.

- Given that even in the first parts of section 2 the authors discussed the role of LN class, I suggest the authors to consider the possibility to modify the organization of the paper in order to firstly report the LN classification.

Thank you for the suggestion. We have rearranged some paragraphs in line with all the suggestions from all the reviewers in order to make the manuscript more easy to follow. However, it our view and the way we think in clinical practice that following the initial evaluation of a patients with SLE we integrate the clinical features to point us towards the predominant type of renal involvement and construct thereafter our management algorithm. There are situations, such as TMA, in which the biopsy may be rarely performed in the acute setting and the diagnosis is maybe a clinical one. Accordingly, treatment may be started before performing a kidney biopsy. After the initial suspicion, there is a need to delineate the indication of kidney biopsy and define the way to integrate the initial histological information into clinical practice. This was our review algorithm process and we delineated further subchapters to emphasize these aspects better.

- Please review the paper in order to define the term the first time they appear in the text (eg: page 1, line 42 - LN; page 2, line 49 - ESRD; page 4, line 90 - GN; page 5, line 149 - IS; page 5, line 172 - eGFR; page 5, line 180 EULAR/ERA-EDTA; page 5, line 182 - KDIGO).

We have adjusted the abbreviations according to your suggestions.

- Please report the legend for table I, II, III, IV and V.

We have added the legend for all the tables. Thank you for the observation.

Round 2

Reviewer 2 Report

Dear Authors,

Thank you for addressing my concerns in such a professional and timely manner. I believe that the changes you made further highlight the value of your work compared to the initial version of the manuscript.

I have no further comments.